# Bayesian inference as iterated random functions with applications to sequential inference in graphical models

**Arash A. Amini**
Department of Statistics
University of Michigan
Ann Arbor, Michigan 48109
aaamini@umich.edu

**XuanLong Nguyen**
Department of Statistics
University of Michigan
Ann Arbor, Michigan 48109
xuanlong@umich.edu

## Abstract

We propose a general formalism of iterated random functions with semigroup property, under which exact and approximate Bayesian posterior updates can be viewed as specific instances. A convergence theory for iterated random functions is presented. As an application of the general theory we analyze convergence behaviors of exact and approximate message-passing algorithms that arise in a sequential change point detection problem formulated via a latent variable directed graphical model. The sequential inference algorithm and its supporting theory are illustrated by simulated examples.

## 1 Introduction

The sequential posterior updates play a central role in many Bayesian inference procedures. As an example, in Bayesian inference one is interested in the posterior probability of variables of interest given the data observed sequentially up to a given time point. As a more specific example which provides the motivation for this work, in a sequential change point detection problem [1], the key quantity is the posterior probability that a change has occurred given the data observed up to present time. When the underlying probability model is complex, e.g., a large-scale graphical model, the calculation of such quantities in a fast and online manner is a formidable challenge. In such situations approximate inference methods are required – for graphical models, message-passing variational inference algorithms present a viable option [2, 3].

In this paper we propose to treat Bayesian inference in a complex model as a specific instance of an abstract system of iterated random functions (IRF), a concept that originally arises in the study of Markov chains and stochastic systems [4]. The key technical property of the proposed IRF formalism that enables the connection to Bayesian inference under conditionally independent sampling is the *semigroup* property, which shall be defined shortly in the sequel. It turns out that most exact and approximate Bayesian inference algorithms may be viewed as specific instances of an IRF system. The goal of this paper is to present a general convergence theory for the IRF with semigroup property. The theory is then applied to the analysis of exact and approximate message-passing inference algorithms, which arise in the context of distributed sequential change point problems using latent variable and directed graphical model as the underlying modeling framework.

We wish to note a growing literature on message-passing and sequential inference based on graphical modeling [5, 6, 7, 8]. On the other hand, convergence and error analysis of message-passing algorithms in graphical models is quite rare and challenging, especially for approximate algorithms, and they are typically confined to the specific form of belief propagation (sum-product) algorithm [9, 10, 11]. To the best of our knowledge, there is no existing work on the analysis of message-passing inference algorithms for calculating conditional (posterior) probabilities for latent random

variables present in a graphical model. While such an analysis is a byproduct of this work, the viewpoint we put forward here that equates Bayesian posterior updates to a system of iterated random functions with semigroup property seems to be new and may be of general interest.

The paper is organized as follows. In Sections 2– 3, we introduce the general IRF system and provide our main result on its convergence. The proof is deferred to Section 5. As an example of the application of the result, we will provide a convergence analysis for an approximate sequential inference algorithm for the problem of multiple change point detection using graphical models. The problem setup and the results are discussed in Section 4.

## 2 Bayesian posterior updates as iterated random functions

In this paper we shall restrict ourselves to multivariate distributions of binary random variables. To describe the general iteration, let $\mathcal{P}_d := \mathcal{P}(\{0,1\}^d)$ be the space of probability measures on $\{0,1\}^d$. The iteration under consideration recursively produces a random sequence of elements of $\mathcal{P}_d$, starting from some initial value. We think of $\mathcal{P}_d$ as a subset of $\mathbb{R}^{2^d}$ equipped with the $\ell_1$ norm (that is, the total variation norm for discrete probability measures). To simplify, let $m := 2^d$, and for $x \in \mathcal{P}_d$, index its coordinates as $x = (x^0, \ldots, x^{m-1})$. For $\boldsymbol{\theta} \in \mathbb{R}_+^m$, consider the function $q_{\boldsymbol{\theta}} : \mathcal{P}_d \to \mathcal{P}_d$, defined by

$$q_{\boldsymbol{\theta}}(x) := \frac{x \odot \boldsymbol{\theta}}{x^T \boldsymbol{\theta}} \tag{1}$$

where $x^T \boldsymbol{\theta} = \sum_i x^i \boldsymbol{\theta}^i$ is the usual inner product on $\mathbb{R}^m$ and $x \odot \boldsymbol{\theta}$ is pointwise multiplication with coordinates $[x \odot \boldsymbol{\theta}]^i := x^i \boldsymbol{\theta}^i$, for $i = 0, 1, \ldots, m-1$. This function models the prior-to-posterior update according to the Bayes rule. One can think of $\boldsymbol{\theta}$ as the likelihood and $x$ as the prior distribution (or the posterior in the previous stage) and $q_{\boldsymbol{\theta}}(x)$ as the (new) posterior based on the two. The division by $x^T \boldsymbol{\theta}$ can be thought of as the division by the marginal to make a valid probability vector. (See Example 1 below.)

We consider the following general iteration

$$\begin{aligned} Q_n(x) &= q_{\boldsymbol{\theta}_n}(T(Q_{n-1}(x))), \quad n \geq 1, \\ Q_0(x) &= x, \end{aligned} \tag{2}$$

for some deterministic operator $T : \mathcal{P}_d \to \mathcal{P}_d$ and an i.i.d. random sequence $\{\boldsymbol{\theta}_n\}_{n \geq 1} \subset \mathbb{R}_+^m$. By changing operator $T$, one obtains different iterative algorithms.

Our goal is to find sufficient conditions on $T$ and $\{\boldsymbol{\theta}_n\}$ for the convergence of the iteration to an extreme point of $\mathcal{P}_d$, which without loss of generality is taken to be $e^{(0)} := (1, 0, 0, \ldots, 0)$. Standard techniques for proving the convergence of iterated random functions are usually based on showing some averaged-sense contraction property for the iteration function [4, 12, 13, 14], which in our case is $q_{\boldsymbol{\theta}_n}(T(\cdot))$. See [15] for a recent survey. These techniques are not applicable to our problem since $q_{\boldsymbol{\theta}_n}$ is not in general Lipschitz, in any suitable sense, precluding $q_{\boldsymbol{\theta}_n}(T(\cdot))$ from satisfying the aforementioned conditions.

Instead, the functions $\{q_{\boldsymbol{\theta}_n}\}$ have another property which can be exploited to prove convergence; namely, they form a semi-group under pointwise multiplication,

$$q_{\boldsymbol{\theta} \odot \boldsymbol{\theta}'} = q_{\boldsymbol{\theta}} \circ q_{\boldsymbol{\theta}'}, \quad \boldsymbol{\theta}, \boldsymbol{\theta}' \in \mathbb{R}_+^m, \tag{3}$$

where $\circ$ denotes the composition of functions. If $T$ is the identity, this property allows us to write $Q_n(x) = q_{\odot_{i=1}^n \boldsymbol{\theta}_i}(x)$ — this is nothing but the Bayesian posterior update equation, under conditionally independent sampling, while modifying $T$ results in an approximate Bayesian inference procedure. Since after suitable normalization, $\odot_{i=1}^n \boldsymbol{\theta}_i$ concentrates around a deterministic quantity, by the i.i.d. assumption on $\{\boldsymbol{\theta}_i\}$, this representation helps in determining the limit of $\{Q_n(x)\}$. The main result of this paper, summarized in Theorem 1, is that the same conclusions can be extended to general Lipschitz maps $T$ having the desired fixed point.

# 3 General convergence theory

Consider a sequence $\{\boldsymbol{\theta}_n\}_{n \geq 1} \subset \mathbb{R}_+^m$ of i.i.d. random elements, where $m = 2^d$. Let $\boldsymbol{\theta}_n = (\boldsymbol{\theta}_n^0, \boldsymbol{\theta}_n^1, \ldots, \boldsymbol{\theta}_n^{m-1})$ with $\boldsymbol{\theta}_n^0 = 1$ for all $n$, and

$$\boldsymbol{\theta}_n^* := \max_{i=1,2,\ldots,m-1} \boldsymbol{\theta}_n^i. \tag{4}$$

The normalization $\boldsymbol{\theta}_n^0 = 1$ is convenient for showing convergence to $\boldsymbol{e}^{(0)}$. This is without loss of generality, since $q_{\boldsymbol{\theta}}$ is invariant to scaling of $\boldsymbol{\theta}$, that is $q_{\boldsymbol{\theta}} = q_{\beta\boldsymbol{\theta}}$ for any $\beta > 0$.

Assume the sequence $\{\log \boldsymbol{\theta}_n^*\}$ to be i.i.d. sub-Gaussian with mean $\leq -I_* < 0$ and sub-Gaussian norm $\leq \sigma_* \in (0, \infty)$. The sub-Gaussian norm in can be taken to be the $\psi_2$ Orlicz norm (cf. [16, Section 2.2]), which we denote by $\|\cdot\|_{\psi_2}$. By definition $\|Y\|_{\psi_2} := \inf\{C > 0 : \mathbb{E}\psi_2(|Y|/C) \leq 1\}$ where $\psi_2(x) := e^{x^2} - 1$.

Let $\|\cdot\|$ denote the $\ell_1$ norm on $\mathbb{R}^m$. Consider the sequence $\{Q_n(x)\}_{n \geq 0}$ defined in (2) based on $\{\boldsymbol{\theta}_n\}$ as above, an initial point $x = (x^0, \ldots, x^{m-1}) \in \mathcal{P}_d$ and a Lipschitz map $T : \mathcal{P}_d \to \mathcal{P}_d$. Let $\mathrm{Lip}_T$ denote the Lipschitz constant of $T$, that is $\mathrm{Lip}_T := \sup_{x \neq y} \|T(x) - T(y)\|/\|x - y\|$.

Our main result regarding iteration (2) is the following.

**Theorem 1.** *Assume that $L := \mathrm{Lip}_T \leq 1$ and that $\boldsymbol{e}^{(0)}$ is a fixed point of $T$. Then, for all $n \geq 0$, and $\varepsilon > 0$,*

$$\|Q_n(x) - \boldsymbol{e}^{(0)}\| \leq 2\frac{1 - x^0}{x^0}\left(Le^{-I_*+\varepsilon}\right)^n \tag{5}$$

*with probability at least $1 - \exp(-c\,n\varepsilon^2/\sigma_*^2)$, for some absolute constant $c > 0$.*

The proof of Theorem 1 is outlined in Section 5. Our main application of the theorem will be to the study of convergence of stopping rules for a distributed multiple change point problem endowed with latent variable graphical models. Before stating that problem, let us consider the classical (single) change point problem first, and show how the theorem can be applied to analyze the convergence of the optimal Bayes rule.

**Example 1.** *In the classical Bayesian change point problem [1], one observes a sequence $\{X^1, X^2, X^3 \ldots\}$ of independent data points whose distributions change at some random time $\lambda$. More precisely, given $\lambda = k$, $X^1, X^2, \ldots, X^{k-1}$ are distributed according to $g$, and $X^{k+1}, X^{k+2}, \ldots$ according to $f$. Here, $f$ and $g$ are densities with respect to some underlying measure. One also assumes a prior $\pi$ on $\lambda$, usually taken to be geometric. The goal is to find a stopping rule $\tau$ which can predict $\lambda$ based on the data points observed so far. It is well-known that a rule based on thresholding the posterior probability of $\lambda$ is optimal (in a Neyman-Pearson sense). To be more specific, let $\mathbf{X}^n := (X^1, X^2, \ldots, X^n)$ collect the data up to time $n$ and let $\gamma^n[n] := \mathbb{P}(\lambda \leq n | \mathbf{X}^n)$ be the posterior probability of $\lambda$ having occurred before (or at) time $n$. Then, the Shiryayev rule*

$$\tau := \inf\{n \in \mathbb{N} : \gamma^n[n] \geq 1 - \alpha\} \tag{6}$$

*is known to asymptotically have the least expected delay, among all stopping rules with false alarm probability bounded by $\alpha$.*

Theorem 1 provides a way to quantify how fast the posterior $\gamma^n[n]$ approaches 1, once the change point has occurred, hence providing an estimate of the detection delay, even for finite number of samples. We should note that our approach here is somewhat independent of the classical techniques normally used for analyzing stopping rule (6). To cast the problem in the general framework of (2), let us introduce the binary variable $Z^n := 1\{\lambda \leq n\}$, where $1\{\cdot\}$ denotes the indicator of an event. Let $Q_n$ be the (random) distribution of $Z^n$ given $\mathbf{X}^n$, in other words,

$$Q_n := \left(\mathbb{P}(Z^n = 1|\mathbf{X}^n), \mathbb{P}(Z^n = 0|\mathbf{X}^n)\right).$$

Since $\gamma^n[n] = \mathbb{P}(Z = 1|\mathbf{X}^n)$, convergence of $\gamma^n[n]$ to 1 is equivalent to the convergence of $Q_n$ to $\boldsymbol{e}^{(0)} = (1, 0)$. We have

$$P(Z^n|\mathbf{X}^n) \propto_{Z^n} P(Z^n, X^n|\mathbf{X}^{n-1}) = P(X^n|Z^n)P(Z^n|\mathbf{X}^{n-1}). \tag{7}$$

Note that $P(X^n|Z^n = 1) = f(X^n)$ and $P(X^n|Z^n = 0) = g(X^n)$. Let $\boldsymbol{\theta}_n := \left(1, \frac{g(X^n)}{f(X^n)}\right)$ and

$$\mathcal{R}_{n-1} := \left(\mathbb{P}(Z^n = 1|\mathbf{X}^{n-1}), \mathbb{P}(Z^n = 0|\mathbf{X}^{n-1})\right).$$

Then, (7) implies that $Q_n$ can be obtained by pointwise multiplication of $\mathcal{R}_{n-1}$ by $f(X^n)\boldsymbol{\theta}_n$ and normalization to make a probability vector. Alternatively, we can multiply by $\boldsymbol{\theta}_n$, since the procedure is scale-invariant, that is, $Q_n = q_{\boldsymbol{\theta}_n}(\mathcal{R}_{n-1})$ using definition (1). It remains to express $\mathcal{R}_{n-1}$ in terms of $Q_{n-1}$. This can be done by using the Bayes rule and the fact that $P(\mathbf{X}^{n-1}|\lambda = k)$ is the same for $k \in \{n, n+1, \dots\}$. In particular, after some algebra (see [17]), one arrives at

$$\gamma^{n-1}[n] = \frac{\pi(n)}{\pi[n-1]^c} + \frac{\pi[n]^c}{\pi[n-1]^c}\gamma^{n-1}[n-1], \tag{8}$$

where $\gamma^k[n] := \mathbb{P}(\lambda \le n|\mathbf{X}^k)$, $\pi(n)$ is the prior on $\lambda$ evaluated at time $n$, and $\pi[k]^c := \sum_{i=k+1}^{\infty} \pi(i)$. For the geometric prior with parameter $\rho \in [0,1]$, we have $\pi(n) := (1-\rho)^{n-1}\rho$ and $\pi[k]^c = \rho^k$. The above recursion then simplifies to $\gamma^{n-1}[n] = \rho + (1-\rho)\gamma^{n-1}[n-1]$. Expressing in terms of $\mathcal{R}_{n-1}$ and $Q_{n-1}$, the recursion reads

$$\mathcal{R}_{n-1} = T(Q_{n-1}), \quad \text{where } T\left(\begin{pmatrix} x_1 \\ x_0 \end{pmatrix}\right) = \rho\begin{pmatrix} 1 \\ 0 \end{pmatrix} + (1-\rho)\begin{pmatrix} x_1 \\ x_0 \end{pmatrix}.$$

In other words, $T(x) = \rho\boldsymbol{e}^{(0)} + (1-\rho)x$ for $x \in \mathcal{P}_2$.

Thus, we have shown that an iterative algorithm for computing $\gamma^n[n]$ (hence determining rule (6)), can be expressed in the form of (2) for appropriate choices of $\{\boldsymbol{\theta}_n\}$ and operator $T$. Note that $T$ in this case is Lipschitz with constant $1 - \rho$ which is always guaranteed to be $\le 1$.

We can now use Theorem 1 to analyze the convergence of $\gamma^n[n]$. Let us condition on $\lambda = k + 1$, that is, we assume that the change point has occurred at time $k + 1$. Then, the sequence $\{X^n\}_{n \ge k+1}$ is distributed according to $f$, and we have $\mathbb{E}\boldsymbol{\theta}_n^* = \int f \log \frac{g}{f} = -I$, where $I$ is the KL divergence between densities $f$ and $g$. Noting that $\|Q_n - \boldsymbol{e}^{(0)}\| = 2(1 - \gamma^n[n])$, we immediately obtain the following corollary.

**Corollary 1.** *Consider Example 1 and assume that* $\log(g(X)/f(X))$, *where* $X \sim f$, *is sub-Gaussian with sub-Gaussian norm* $\le \sigma$. *Let* $I := \int f \log \frac{f}{g}$. *Then, conditioned on* $\lambda = k + 1$, *we have for* $n \ge 1$,

$$\left|\gamma^{n+k}[n+k] - 1\right| \le \left[(1-\rho)e^{-I+\varepsilon}\right]^n \left(\frac{1}{\gamma^k[k]} - 1\right)$$

*with probability at least* $1 - \exp(-c\,n\varepsilon^2/\sigma^2)$.

## 4 Multiple change point problem via latent variable graphical models

We now turn to our main application for Theorem 1, in the context of a multiple change point problem. In [18], graphical model formalism is used to extend the classical change point problem (cf. Example 1) to cases where multiple distributed latent change points are present. Throughout this section, we will use this setup which we now briefly sketch.

One starts with a network $G = (V, E)$ of $d$ sensors or nodes, each associated with a change point $\lambda_j$. Each node $j$ observes a private sequence of measurements $\mathbf{X}_j = (X_j^1, X_j^2, \dots)$ which undergoes a change in distribution at time $\lambda_j$, that is,

$$X_j^1, X_j^2, \dots, X_j^{k-1} \mid \lambda_j = k \overset{iid}{\sim} g_j, \qquad X_j^k, X_j^{k+1}, \dots \mid \lambda_j = k \overset{iid}{\sim} f_j,$$

for densities $g_j$ and $f_j$ (w.r.t. some underlying measure). Each connected pair of nodes share an additional sequence of measurements. For example, if nodes $s_1$ and $s_2$ are connected, that is, $e = (s_1, s_2) \in E$, then they both observe $\mathbf{X}_e = (X_e^1, X_e^2, \dots)$. The shared sequence undergoes a change in distribution at some point depending on $\lambda_{s_1}$ and $\lambda_{s_2}$. More specifically, it is assumed that the earlier of the two change points causes a change in the shared sequence, that is, the distribution of $\mathbf{X}_e$ conditioned on $(\lambda_{s_1}, \lambda_{s_2})$ only depends on $\lambda_e := \lambda_{s_1} \wedge \lambda_{s_2}$, the minimum of the two, i.e.,

$$X_e^1, X_e^2, \dots, X_e^k \mid \lambda_e = k \overset{iid}{\sim} g_e, \qquad X_e^{k+1}, X_e^{k+2}, \dots \mid \lambda_e = k \overset{iid}{\sim} f_e.$$

Letting $\lambda_* := \{\lambda_j\}_{j \in V}$ and $\mathbf{X}_*^n = \{\mathbf{X}_j^n, \mathbf{X}_e^n\}_{j \in V, e \in E}$, we can write the joint density of all random variables as

$$P(\lambda_*, \mathbf{X}_*^n) = \prod_{j \in V} \pi_j(\lambda_j) \prod_{j \in V} P(\mathbf{X}_j^n | \lambda_j) \prod_{e \in E} P(\mathbf{X}_e^n | \lambda_{s_1}, \lambda_{s_2}). \tag{9}$$

where $\pi_j$ is the prior on $\lambda_j$, which we assume to be geometric with parameter $\rho_j$. Network $G$ induces a graphical model [2] which encodes the factorization (9) of the joint density. (cf. Fig. 1)

Suppose now that each node $j$ wants to detect its change point $\lambda_j$, with minimum expected delay, while maintaining a false alarm probability at most $\alpha$. Inspired by the classical change point problem, one is interested in computing the posterior probability that the change point has occurred up to now, that is,

$$\gamma_j^n[n] := \mathbb{P}(\lambda_j \leq n \mid \mathbf{X}_*^n). \tag{10}$$

The difference with the classical setting is the conditioning is done on all the data in the network (up to time $n$). It is easy to verify that the natural stopping rule

$$\tau_j = \inf\{n \in \mathbb{N} : \ \gamma_j^n[n] \geq 1 - \alpha\}$$

satisfy the false alarm constraint. It has also been shown that this rule is asymptotically optimal in terms of expected detection delay. Moreover, an algorithm based on the well-known sum-product [2] has been proposed, which allows the nodes to compute their posterior probabilities 10 by message-passing. The algorithm is exact when $G$ is a tree, and scales linearly in the number of nodes. More precisely, at time $n$, the computational complexity is $O(nd)$. The drawback is the linear dependence on $n$, which makes the algorithm practically infeasible if the change points model rare events (where $n$ could grow large before detecting the change.)

In the next section, we propose an approximate message passing algorithm which has computational complexity $O(d)$, at each time step. This circumvents the drawback of the exact algorithm and allows for indefinite run times. We then show how the theory developed in Section 3 can be used to provide convergence guarantees for this approximate algorithm, as well as the exact one.

### 4.1 Fast approximate message-passing (MP)

We now turn to an approximate message-passing algorithm which, at each time step, has computational complexity $O(d)$. The derivation is similar to that used for the iterative algorithm in Example 1. Let us define binary variables

$$Z_j^n = 1\{\lambda_j \leq n\}, \quad Z_*^n = (Z_1^n, \ldots, Z_d^n). \tag{11}$$

The idea is to compute $P(Z_*^n | \mathbf{X}_*^n)$ recursively based on $P(Z_*^{n-1} | \mathbf{X}_*^{n-1})$. By Bayes rule, $P(Z_*^n | \mathbf{X}_*^n)$ is proportional in $Z_*^n$ to $P(Z_*^n, X_*^n | \mathbf{X}_*^{n-1}) = P(X_*^n | Z_*^n) P(Z_*^n | \mathbf{X}_*^{n-1})$, hence

$$P(Z_*^n | \mathbf{X}_*^n) \propto_{Z_*^n} \left[ \prod_{j \in V} P(X_j^n | Z_j^n) \prod_{\{i,j\} \in E} P(X_{ij}^n | Z_i^n, Z_j^n) \right] P(Z_*^n | \mathbf{X}_*^{n-1}), \tag{12}$$

where we have used the fact that given $Z_*^n$, $X_*^n$ is independent of $\mathbf{X}_*^{n-1}$. To simplify notation, let us extend the edge set to $\widetilde{E} := E \cup \{\{j\} : j \in V\}$. This allows us to treat the private data of node $j$, i.e., $\mathbf{X}_j$, as shared data of a self-loop in the extended graph $(V, \widetilde{E})$. Let $u_e(z; \xi) := [g_e(\xi)]^{1-z} [f_e(\xi)]^z$ for $e \in \widetilde{E}$, $z \in \{0, 1\}$. Then, for $i \neq j$,

$$P(X_j^n | Z_j^n) = u_j(Z_j^n; X_j^n), \quad P(X_{ij}^n | Z_i^n, Z_j^n) = u_{ij}(Z_i^n \vee Z_j^n; X_{ij}^n). \tag{13}$$

It remains to express $P(Z_*^n | \mathbf{X}_*^{n-1})$ in terms of $P(Z_*^{n-1} | \mathbf{X}_*^{n-1})$. It is possible to do this, exactly, at a cost of $O(2^{|V|})$. For brevity, we omit the exact expression. (See Lemma 1 for some details.) We term the algorithm that employs the exact relationship, the "exact algorithm".

In practice, however, the exponential complexity makes the exact recursion of little use for large networks. To obtain a fast algorithm (i.e., $O(\mathrm{poly}(d))$), we instead take a mean-field type approximation:

$$P(Z_*^n | \mathbf{X}_*^{n-1}) \approx \prod_{j \in V} P(Z_j^n | \mathbf{X}_*^{n-1}) = \prod_{j \in V} \nu(Z_j^n; \gamma_j^{n-1}[n]), \tag{14}$$

where $\nu(z;\beta) := \beta^z(1-\beta)^{1-z}$. That is, we approximate a multivariate distribution by the product of its marginals. By an argument similar to that used to derive (8), we can obtain a recursion for the marginals,

$$\gamma_j^{n-1}[n] = \frac{\pi_j(n)}{\pi_j[n-1]^c} + \frac{\pi_j[n]^c}{\pi_j[n-1]^c}\gamma_j^{n-1}[n-1], \tag{15}$$

where we have used the notation introduced earlier in (8). Thus, at time $n$, the RHS of (14) is known based on values computed at time $n-1$ (with initial value $\gamma_j^0[0] = 0, j \in V$). Inserting this RHS into (12) in place of $P(Z_*^n|\mathbf{X}_*^{n-1})$, we obtain a graphical model in variables $Z_*^n$ (instead of $\lambda_*$) which has the same form as (9) with $\nu(Z_j^n; \gamma_j^{n-1}[n])$ playing the role of the prior $\pi(\lambda_j)$.

In order to obtain the marginals $\gamma_j^n[n] = P(Z_j^n = 1|\mathbf{X}_*^n)$ and $\gamma_{ij}^n[n]$ with respect to the approximate version of the joint distribution $P(Z_*^n, X_*^n|\mathbf{X}_*^{n-1})$, we need to marginalize out the latent variables $Z_j^n$'s, for which a standard sum-product algorithm can be applied (see [2, 3, 18]). The message update equations are similar to those in [18]; the difference is that the messages are now binary and do not grow in size with $n$.

## 4.2 Convergence of MP algorithms

We now turn to the analysis of the approximate algorithm introduced in Section 4.1. In particular, we will look at the evolution of $\{\widetilde{P}(Z_*^n|\mathbf{X}_*^n)\}_{n\in\mathbb{N}}$ as a sequence of probability distribution on $\{0,1\}^d$. Here, $\widetilde{P}$ signifies that this sequence is an approximation. In order to make a meaningful comparison, we also look at the algorithm which computes the exact sequence $\{P(Z_*^n|\mathbf{X}_*^n)\}_{n\in\mathbb{N}}$, recursively. As mentioned before, this we will call the "exact algorithm", the details of which are not of concern to us at this point (cf. Prop. 1 for these details.)

Recall that we take $\widetilde{P}(Z_*^n|\mathbf{X}_*^n)$ and $P(Z_*^n|\mathbf{X}_*^n)$, as distributions for $Z_*^n$, to be elements of $\mathcal{P}_d \subset \mathbb{R}^m$. To make this correspondence formal and the notation simplified, we use the symbol $:\equiv$ as follows

$$\widetilde{y}_n :\equiv \widetilde{P}(Z_*^n|\mathbf{X}_*^n), \quad y_n :\equiv P(Z_*^n|\mathbf{X}_*^n) \tag{16}$$

where now $\widetilde{y}_n, y_n \in \mathcal{P}_d$. Note that $\widetilde{y}_n$ and $y_n$ are random elements of $\mathcal{P}_d$, due the randomness of $\mathbf{X}_*^n$. We have the following description.

**Proposition 1.** *The exact and approximate sequences, $\{y_n\}$ and $\{\widetilde{y}_n\}$, follow general iteration* (2) *with the same random sequence $\{\boldsymbol{\theta}_n\}$, but with different deterministic operators $T$, denoted respectively with $T_{\mathrm{ex}}$ and $T_{\mathrm{ap}}$. $T_{\mathrm{ex}}$ is linear and given by a Markov transition kernel. $T_{\mathrm{ap}}$ is a polynomial map of degree $d$. Both maps are Lipschitz and we have*

$$\mathrm{Lip}_{T_{\mathrm{ex}}} \le L_\rho := \Big(1 - \prod_{j=1}^d \rho_j\Big), \quad \mathrm{Lip}_{T_{\mathrm{ap}}} \le K_\rho := \sum_{j=1}^d (1 - \rho_j). \tag{17}$$

Detailed descriptions of the sequence $\{\boldsymbol{\theta}_n\}$ and the operators $T_{\mathrm{ex}}$ and $T_{\mathrm{ap}}$ are given in [17]. As suggested by Theorem 1, a key assumption for the convergence of the approximate algorithm will be $K_\rho \le 1$. In contrast, we always have $L_\rho \le 1$.

Recall that $\{\lambda_j\}$ are the change points and their priors are geometric with parameters $\{\rho_j\}$. We analyze the algorithms, once all the change points have happened. More precisely, we condition on $\mathbb{M}_{n_0} := \{\max_j \lambda_j \le n_0\}$ for some $n_0 \in \mathbb{N}$. Then, one expects the (joint) posterior of $Z_*^n$ to contract to the point $Z_j^\infty = 1$, for all $j \in V$. In the vectorial notation, we expect both $\{\widetilde{y}_n\}$ and $\{y_n\}$ to converge to $e^{(0)}$. Theorem 2 below quantifies this convergence in $\ell_1$ norm (equivalently, total variation for measures).

Recall pre-change and post-change densities $g_e$ and $f_e$, and let $I_e$ denote their KL divergence, that is, $I_e := \int f_e \log(f_e/g_e)$. We will assume that

$$Y_e := \log(g_e(X)/f_e(X)) \quad \text{with} \quad X \sim f_e \tag{18}$$

is sub-Gaussian, for all $e \in \widetilde{E}$, where $\widetilde{E}$ is extended edge notation introduced in Section 4.1. The choice $X \sim f_e$ is in accordance with conditioning on $\mathbb{M}_{n_0}$. Note that $\mathbb{E}Y_e = -I_e < 0$. We define

$$\sigma_{\max} := \max_{e \in \widetilde{E}} \|Y_e\|_{\psi_2}, \quad I_{\min} := \min_{e \in \widetilde{E}} I_e, \quad I_*(\kappa) := I_{\min} - \kappa\, \sigma_{\max}\sqrt{\log D}..$$

where $D := |V| + |E|$. Theorem 1 and Lemma 1 give us the following. (See [17] for the proof.)

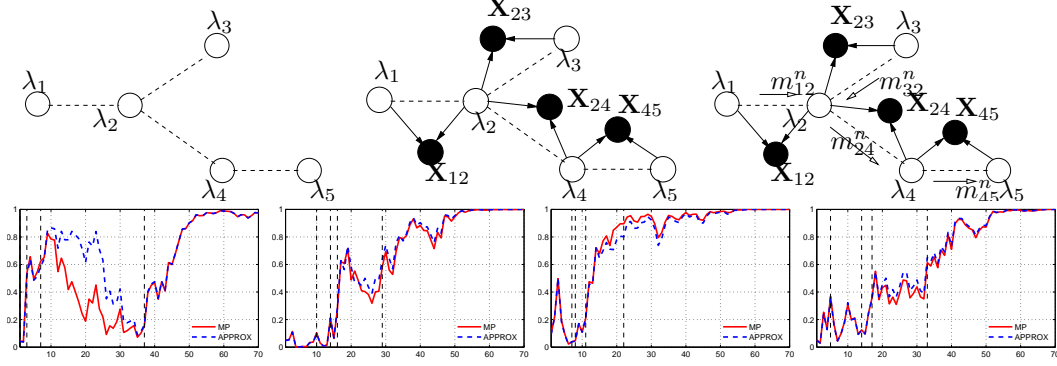

Figure 1: Top row illustrates a network (left), which induces a graphical model (middle). Right panel illustrates one stage of message-passing to compute posterior probabilities $\gamma_j^n[n]$. Bottom row illustrates typical examples of posterior paths, $n \mapsto \gamma_j^n[n]$, obtained by EXACT and approximate (APPROX) message passing, for the subgraph on nodes $\{1, 2, 3, 4\}$. The change points are designated with vertical dashed lines.

**Theorem 2.** *There exists an absolute constant $\kappa > 0$, such that if $I_*(\kappa) > 0$, the exact algorithm converges at least geometrically w.h.p., that is, for all $n \geq 1$,*

$$\|y_{n+n_0} - \boldsymbol{e}^{(0)}\| \leq 2 \frac{1 - y_{n_0}}{y_{n_0}} \left( L_\rho e^{-I_*(\kappa) + \varepsilon} \right)^n \tag{19}$$

*with probability at least $1 - \exp\left[-c\, n\varepsilon^2/(\sigma_{\max}^2 D^2 \log D)\right]$, conditioned on $\mathbb{M}_{n_0}$. If in addition, $K_\rho \leq 1$, the approximate algorithm also converges at least geometrically w.h.p., i.e., for all $n \geq 1$,*

$$\|\widetilde{y}_{n+n_0} - \boldsymbol{e}^{(0)}\| \leq 2 \frac{1 - \widetilde{y}_{n_0}}{\widetilde{y}_{n_0}} \left( K_\rho e^{-I_*(\kappa) + \varepsilon} \right)^n \tag{20}$$

*with the same (conditional) probability as the exact algorithm.*

### 4.3 Simulation results

We present some simulation results to verify the effectiveness of the proposed approximation algorithm in estimating the posterior probabilities $\gamma_j^n[n]$. We consider a star graph on $d = 4$ nodes. This is the subgraph on nodes $\{1, 2, 3, 4\}$ in Fig. 1. Conditioned on the change points $\lambda_*$, all data sequences $\mathbf{X}_*$ are assumed Gaussian with variance 1, pre-change mean 1 and post-change mean zero. All priors are geometric with $\rho_j = 0.1$. We note that higher values of $\rho_j$ yield even faster convergence in the simulations, but we omit these figures due to space constraints. Fig. 1 illustrates typical examples of posterior paths $n \mapsto \gamma_j^n[n]$, for both the exact and approximate MP algorithms. One can observe that the approximate path often closely follows the exact one. In some cases, they might deviate for a while, but as suggested by Theorem 2, they approach one another quickly, once the change points have occurred.

From the theorem and triangle inequality, it follows that under $I_*(\kappa) > 0$ and $K_\rho \leq 1$, $\|y_n - \widetilde{y}_n\|$ converges to zero, at least geometrically w.h.p. This gives some theoretical explanation for the good tracking behavior of approximate algorithm as observed in Fig. 1.

## 5 Proof of Theorem 1

For $x \in \mathbb{R}^m$ (including $\mathcal{P}_d$), we write $x = (x^0, \widetilde{x})$ where $\widetilde{x} = (x^1, \ldots, x^{m-1})$. Recall that $\boldsymbol{e}^{(0)} = (1, 0, \ldots, 0)$ and $\|x\| = \sum_{i=0}^{m-1} |x_i|$. For $x \in \mathcal{P}_d$, we have $1 - x^0 = \|\widetilde{x}\|$, and

$$\|x - \boldsymbol{e}^{(0)}\| = \|(x^0 - 1, \widetilde{x})\| = 1 - x^0 + \|\widetilde{x}\| = 2(1 - x^0). \tag{21}$$

For $\boldsymbol{\theta} = (\boldsymbol{\theta}^0, \widetilde{\boldsymbol{\theta}}) \in \mathbb{R}_+^m$, let

$$\boldsymbol{\theta}^* := \|\widetilde{\boldsymbol{\theta}}\|_\infty = \max_{i=1,\ldots,m-1} \boldsymbol{\theta}^i, \qquad \boldsymbol{\theta}^\dagger := \left( \boldsymbol{\theta}^0, (\boldsymbol{\theta}^* L) \mathbf{1}_{m-1} \right) \in \mathbb{R}_+^m \tag{22}$$

where $\mathbf{1}_{m-1}$ is a vector in $\mathbb{R}^{m-1}$ whose coordinates are all ones. We start by investigating how $\|q_{\boldsymbol{\theta}}(x) - \boldsymbol{e}^{(0)}\|$ varies as a function of $\|x - \boldsymbol{e}^{(0)}\|$.

**Lemma 1.** *For $L \leq 1$, $\boldsymbol{\theta}^* > 0$, and $\boldsymbol{\theta}^0 = 1$,*

$$N := \sup_{\substack{x,y \in \mathcal{P}_d, \\ \|x - \boldsymbol{e}^{(0)}\| \leq L \|y - \boldsymbol{e}^{(0)}\|}} \frac{\|q_{\boldsymbol{\theta}}(x) - \boldsymbol{e}^{(0)}\|}{\|q_{\boldsymbol{\theta}^\dagger}(y) - \boldsymbol{e}^{(0)}\|} = 1; \tag{23}$$

Lemma 1 is proved in [17]. We now proceed to the proof of the theorem. Recall that $T : \mathcal{P}_d \to \mathcal{P}_d$ is an $L$-Lipschitz map, and that $\boldsymbol{e}^{(0)}$ is a fixed point of $T$, that is, $T(\boldsymbol{e}^{(0)}) = \boldsymbol{e}^{(0)}$. It follows that for any $x \in \mathcal{P}_d$, $\|T(x) - \boldsymbol{e}^{(0)}\| \leq L\|x - \boldsymbol{e}^{(0)}\|$. Applying Lemma 1, we get

$$\|q_{\boldsymbol{\theta}}(T(x)) - \boldsymbol{e}^{(0)}\| \leq \|q_{\boldsymbol{\theta}^\dagger}(x) - \boldsymbol{e}^{(0)}\| \tag{24}$$

for $\boldsymbol{\theta} \in \mathbb{R}_+^m$ with $\boldsymbol{\theta}^0 = 1$, and $x \in \mathcal{P}_d$. (This holds even if $\boldsymbol{\theta}^* = 0$ where both sides are zero.)

Recall the sequence $\{\boldsymbol{\theta}_n\}_{n \geq 1}$ used in defining functions $\{Q_n\}$ accroding to (2), and the assumption that $\boldsymbol{\theta}_n^0 = 1$, for all $n \geq 1$. Inequality (24) is key in allowing us to peel operator $T$, and bring successive elements of $\{q_{\boldsymbol{\theta}_n}\}$ together. Then, we can exploit the semi-group property (3) on adjacent elements of $\{q_{\boldsymbol{\theta}_n}\}$.

To see this, for each $\boldsymbol{\theta}_n$, let $\boldsymbol{\theta}_n^*$ and $\boldsymbol{\theta}_n^\dagger$ be defined as in (22). Applying (24) with $x$ replaced with $Q_{n-1}(x)$, and $\boldsymbol{\theta}$ with $\boldsymbol{\theta}_n$, we can write

$$\begin{aligned}
\|Q_n(x) - \boldsymbol{e}^{(0)}\| &\leq \|q_{\boldsymbol{\theta}_n^\dagger}(Q_{n-1}(x)) - \boldsymbol{e}^{(0)}\| \quad \text{(by (24))} \\
&= \|q_{\boldsymbol{\theta}_n^\dagger}(q_{\boldsymbol{\theta}_{n-1}}(T(Q_{n-2}(x)))) - \boldsymbol{e}^{(0)}\| \\
&= \|q_{\boldsymbol{\theta}_n^\dagger \odot \boldsymbol{\theta}_{n-1}}(T(Q_{n-2}(x))) - \boldsymbol{e}^{(0)}\| \quad \text{(by semi-group property (3))}
\end{aligned}$$

We note that $(\boldsymbol{\theta}_n^\dagger \odot \boldsymbol{\theta}_{n-1})^* = L \boldsymbol{\theta}_n^* \boldsymbol{\theta}_{n-1}^*$ and

$$\left(\boldsymbol{\theta}_n^\dagger \odot \boldsymbol{\theta}_{n-1}\right)^\dagger = \left(1, L(\boldsymbol{\theta}_n^\dagger \odot \boldsymbol{\theta}_{n-1})^* \mathbf{1}_{m-1}\right) = \left(1, L^2 \boldsymbol{\theta}_n^* \boldsymbol{\theta}_{n-1}^* \mathbf{1}_{m-1}\right).$$

Here, $*$ and $\dagger$ act on a general vector in the sense of (22). Applying (24) once more, we get

$$\|Q_n(x) - \boldsymbol{e}^{(0)}\| \leq \|q_{(1, L^2 \boldsymbol{\theta}_n^* \boldsymbol{\theta}_{n-1}^* \mathbf{1}_{m-1})}(Q_{n-2}(x)) - \boldsymbol{e}^{(0)}\|.$$

The pattern is clear. Letting $\eta_n := L^n \prod_{k=1}^n \boldsymbol{\theta}_k^*$, we obtain by induction

$$\|Q_n(x) - \boldsymbol{e}^{(0)}\| \leq \|q_{(1, \eta_n \mathbf{1}_{m-1})}(Q_0(x)) - \boldsymbol{e}^{(0)}\|. \tag{25}$$

Recall that $Q_0(x) := x$. Moreover,

$$\|q_{(1, \eta_n \mathbf{1}_{m-1})}(x) - \boldsymbol{e}^{(0)}\| = 2\left(1 - [q_{(1, \eta_n \mathbf{1}_{m-1})}(x)]^0\right) = 2\left(1 - g_{\eta_n}(x^0)\right) \tag{26}$$

where the first inequality is by (21), and the second is easily verified by noting that all the elements of $(1, \eta_n \mathbf{1}_{m-1})$, except the first, are equal. Putting (25) and (26) together with the bound $1 - g_\theta(r) = \frac{\theta(1-r)}{r + \theta(1-r)} \leq \theta \frac{1-r}{r}$, which holds for $\theta > 0$ and $r \in (0, 1]$, we obtain $\|Q_n(x) - \boldsymbol{e}^{(0)}\| \leq 2\eta_n \frac{1-x^0}{x^0}$. By sub-Gaussianity assumption on $\{\log \boldsymbol{\theta}_k^*\}$, we have

$$\mathbb{P}\left(\frac{1}{n} \sum_{k=1}^n \log \boldsymbol{\theta}_k^* - \mathbb{E} \log \boldsymbol{\theta}_1^* > \varepsilon\right) \leq \exp(-c\, n\varepsilon^2/\sigma_*^2), \tag{27}$$

for some absolute constant $c > 0$. (Recall that $\sigma_*$ is an upper bound on the sub-Gaussian norm $\|\log \boldsymbol{\theta}_1^*\|_{\psi_2}$.) On the complement of the event in 27, we have $\prod_{k=1}^n \boldsymbol{\theta}_k^* \leq e^{n(-I_* + \varepsilon)}$, which completes the proof.

**Acknowledgments**

This work was supported in part by NSF grants CCF-1115769 and OCI-1047871.

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
