[Reviews · NeurIPS 2013]

Submitted by Assigned_Reviewer_8

This paper gives a new perspective on the Bayesian posterior updates (BPU) as
a system of iterated random functions (IRF) [4].
By proving a general convergence theorem of IRF with semigroup property,
the convergence of algorithms for change point detection problems [17] are analyzed.
The major contributions are
- to establish a general convergence theory for IRF with semigroup property (Theorem 1),
- to cast existing (exact) algorithms for the classical change point problem (CCPP)
and the multiple change point problem (MCPP) in the IRF framework, and prove their
convergence, based on Theorem 1.
- to propose a fast approximate algorithm for MCPP and prove its convergence,
based on Theorem 1.

I think this is a nice paper with a significant impact.
BPU is clearly a Markov chain, and so it might be natural that
it is analyzed in the IRF framework.
However, the previous convergence results on IRF are not directly applicable to BPU,
because the Lipschitz condition is not satisfied in BPU.
Instead, the authors use semigroup property of BPU,
and establish another type of convergence theory.
I expect that the established theory would have wide range of applications.

Pros:
- A new perspective on BPU as a special case of IRF with semigroup property (3) is given.
- A general convergence theory (Theorem 1) for IRF with semigroup property
is established, which might have wide range of potential applications.
- A fast algorithm (Section 4.1) for MCPP is proposed with its convergence
guarantees (Theorem 2).

Cons:
- The relation between the theory (Theorem 2) and the experiment (Figure 1) is not
clear (see the first one in 'Other comments' below).
- No practical application of the multiple change point detection is introduced
(see the second one in 'Other comments' below).

Quality:
The paper is technically sound, and the claims are supported by theory.

Clarity:
The paper is clearly written, and well-organized.

Originality:
The perspective on BPU as IRF with semigroup property,
and the obtained convergence theorem (Theorem 1) are novel.

Significance:
The found relation between BPU and IRF with semigroup property,
and the established convergence theory could be used by other theoreticians
to analyze algorithms for other problems.

Other comments:
- It would be nice if Eqs.(19) and (20) are somehow expressed in the posterior path
graphs in Figure 1. This would make it clear that the simulation supports Theorem 2.
In Section 4.3, it is said that 'in some cases, they might deviate for a while,
but quickly snap back together once the change points have occurred.'
Readers cannot see this because the change points are not indicated.
- It would be nice if any practical example of MCCP is introduced
(I could not find any also in [17]).
Especially, I wonder in what applications the rule
lambda_e = min(lambda_s1, lambda_s2) is appropriate.
To me, it seems more natural to assume that the distribution of X_e
changes both at lambda_s1 and at lambda_s2.
Moreover, if I understand correctly, Theorem 2 guarantees the convergence
after all the sensors are broken.
I suppose people are more interested in the convergence after any of the sensors
is broken, or the sensor on which a user focuses is broken.
- Perhaps, some thetas in and around Eq.(4) should not be bold-faced.
- In Line 225, 'probabilities 10 by message-passing' looks corrupted.
Is 10 a citation?
- In the last line in Section 5, parentheses for Eq.(29) are missing.


Summary: A nice paper that gives a new perspective on the Bayesian posterior updates
as a system of iterated random functions with semigroup property,
and establishes a general convergence theory for the system.
The used techniques and obtained theorems might have wide range
of potential applications.



Submitted by Assigned_Reviewer_9

The paper presents an interesting interpretation of Bayesian inference
as as a system of iterated random functions, with a particular focus on
Bayesian changepoint detection. This interpretation allows for
convergence analysis of message passing algorithms which is of wide
interest. The general approach for convergence analysis is presented
with focus on providing details of the proof technique.

I found the interpretation of Bayesian inference as iterated random
functions interesting and think others will as well. It was interesting
to see that we can derive algorithms that we can show to have geometric
convergence. Overall I think the paper would be improved if there was
more of a discussion and focus on the implications of this theory and
the insights it gives us into obtaining efficient algorithms. Here there
was a specific focus on changepoint detection: was this special for some
reason. What other model classes can we extend this to and for which it is reasonable
to assume the conditions will hold. The simulation section can be improved
to be more clear on the setup, indicating where the changepoints are
and connecting the observed with theoretical convergence over n.
Summary: An interesting interpretation of the paper showing convergence analysis for message passing algorithms. Can be improved by looking into accessibility for many not familiar with the tools used here and the settings in which such tools should be used.

Submitted by Assigned_Reviewer_10

This paper presents iterated random functions and discusses the theory of its convergence. This theory is then used to analyze the convergence of message passing algorithms for Bayesian inference.

Quality: Excellent.

Clarity: Good.

This paper is original and makes significant contributions.

I should admit that I haven't gone through the proofs in complete details, but at the first pass it looks ok to me. I will surely be spending next few days checking things more carefully.

Some questions:
1) What are typical value of -I+epsilon? Is it always negative? If not, under what condition will it be.

2) Can these results be used to explain the convergence for other graphical models where convergence is known, for example, BP on a tree?
Summary: To my knowledge, there is very little work done on the analysis of message-passing algorithm, and hence any such attempts are very useful. This paper uses iterated random functions to analyze the convergence of message passing, which is very interesting. I hope that the presented approach can be generalized to other graphical models, to better understand the convergence of message passing algorithms.
Author Feedback

Author rebuttal: We would like to thank the reviewers for their positive and encouraging comments. Below, we try to answer some of the questions raised.

------------
(Reviewer 1)

- "What are typical values of -I+$\eps$? Is it always negative? If not, under what condition will it be."

We can always choose $\eps$ to make $-I +\eps$ negative as long as the information $I$ is strictly positive, which is a very mild condition. (Consider for example $\eps = I/2$.)

- "Can these results be used to explain the convergence for other graphical models where convergence is known, for example, BP on a tree?"

This is certainly an interesting direction for future extensions. As of now, we do not know how the arguments can be adapted to BP, but we are working on it.

------------
(Reviewer 2)

- "It would be nice if Eqs.(19) and (20) are somehow expressed in the posterior path graphs in Figure 1. This would make it clear that the simulation supports Theorem 2. In Section 4.3, it is said that 'in some cases, they might deviate for a while, but quickly snap back together once the change points have occurred.' Readers cannot see this because the change points are not indicated."

Thanks for the suggestion. We agree and will add markers to indicate the change points.


- "It would be nice if any practical example of MCCP is introduced (I could not find any also in [17]). Especially, I wonder in what applications the rule $\lambda_e = \min(\lambda_{s_1}, \lambda_{s_2})$ is appropriate. To me, it seems more natural to assume that the distribution of $X_e$ changes both at $\lambda_{s_1}$ and at $\lambda_{s_2}$."

The minimum rule is perhaps the simplest form the shared information can take. It simplifies theoretical computations (e.g. finding the $T$ operator and bounding its Lipschitz constant) and can still serve as a first approximation to more realistic situations you have in mind. (Roughly speaking, a shared sequence that changes in response to both change points has more information than the one which responds only to the minimum. So, if the simple model exhibits convergence, we conjecture that the more complex model will do so, at a rate at least as fast as the simple model.)

- "Moreover, if I understand correctly, Theorem 2 guarantees the convergence after all the sensors are broken. I suppose people are more interested in the convergence after any of the sensors is broken, or the sensor on which a user focuses is broken."

Because, we are looking at the convergence of the joint posterior, that is the appropriate regime to look at. Deriving weaker conditions under which a certain marginal of the joint distribution converges is an interesting direction for future investigations.

------------
(Reviewer 3)

-"Here there was a specific focus on changepoint detection: was this special for some reason. What other model classes can we extend this to and for which it is reasonable to assume the conditions will hold?"

We are presently working on extending the theory to the general metric context (e.g., Bayesian models with continuous variable). We are also trying to understand how much of the theory carries to a general belief propagation algorithm (BP).

- "The simulation section can be improved to be more clear on the setup, indicating where the changepoints are and connecting the observed with theoretical convergence over $n$."

This was also pointed out by reviewer 2 and we agree. We will indicate the change points on the plots and comment on the connection with theory.

-------------
We again thanks the reviewers for the thoughtful comments.